# Towards Sustainable Road Pavements: Sound Absorption in Rubber-Modified Asphalt Mixtures

Freddy Richard Apaza [1,*], Víctoriano Fernández Vázquez [2], Santiago Expósito Paje [2], Federico Gulisano [1], Valerio Gagliardi [3], Leticia Saiz Rodríguez [4] and Juan Gallego Medina [1]

[1] Departamento de Ingeniería del Transporte, Territorio y Urbanismo, Universidad Politécnica de Madrid, C/Profesor Aranguren 3, 28040 Madrid, Spain; federico.gulisano@upm.es (F.G.); juan.gallego@upm.es (J.G.M.)

[2] Laboratory of Acoustics Applied to Civil Engineering, University of Castilla—La Mancha, Avda. Camilo José Cela s/n, 13071 Ciudad Real, Spain; victoriano.fernandez@uclm.es (V.F.V.); santiago.exposito@uclm.es (S.E.P.)

[3] Department of Civil, Computer Science and Aeronautical Technologies Engineering, Roma Tre University, Via Vito Volterra 62, 00146 Rome, Italy; valerio.gagliardi@uniroma3.it (V.G.)

[4] Signus Ecovalor S.L. Calle Caleruega, 102, 5°, 28033 Madrid, Spain; lsaiz@signus.es (L.S.R.)

\* Correspondence: fr.aapaza@upm.es

**Abstract:** In the last decade, various asphalt paving materials have undergone investigation for sound attenuation purposes. This research aims to delve into the innovative design of sustainable road pavements by examining sound absorption in rubber-modified asphalt mixtures. More specifically, the impact of alternative sustainable materials on the sound absorption of asphalt mixtures across different temperatures, precisely crumb rubber (CR) derived from recycling of end-of-life tires, was investigated. The acoustic coefficient and its Gaussian fit parameters (Peak, BandWidth, and Area Under the Curve) were evaluated. Five different types of asphalt mixtures were studied, encompassing dense, discontinuous, and open mixtures with 0%, 0.75%, and 1.50% CR incorporated through the dry process (DP). The results of sound absorption indicated a slight influence of crumb rubber at temperatures ranging from 10 °C to 60 °C, particularly in mixtures with high void content. On the other hand, as expected, the void content proved to be highly correlated with sound absorption. These findings facilitated the establishment of predictive models that correlate acoustic absorption spectra with the characteristics of asphalt mixtures. As a result, these models will be valuable in the design of the next generation of sound-absorbing pavements.

**Keywords:** asphalt pavements; crumb rubber; sound absorption; impedance tube; sustainable road pavements; acoustic absorption spectra; rubber-modified asphalt mixtures

## 1. Introduction

Noise pollution is one of the major environmental problems affecting populations around the world. In the last decade, the increase in traffic has led to rising levels of noise pollution [1–3]. This issue poses significant health problems for people living near roads.

To reduce traffic noise to acceptable thresholds, "sound-absorbing pavements" could become one of the most effective alternatives. The majority of transportation modes contribute to increasing noise pollution. This is particularly true for vehicle traffic on asphalt pavements which predominantly accounts for elevated noise levels in urban environments. The mechanisms that dominate noise generation and propagation are influenced by vibrations of moving vehicles [4]. Researchers have investigated the interaction between tires and pavement at speeds exceeding 40 km/h [5], demonstrating that it is the main source of noise as opposed to aerodynamic and mechanical noise.

More specifically, the physical properties of tires and pavements are factors that influence the generation of rolling noise. In recent decades, an extensive literature has

been focused on tire characteristics such as tread shape, carcass, and stiffness patterns and their effects on noise generation [6–8]. On the other hand, the impact of the physico-mechanical properties of pavements on traffic noise reduction has been investigated, as well as the effects related to surface texture, maximum aggregate size, layer thickness, drainage capacity of the asphalt pavement [9], and other material properties, such as stiffness [10,11] and the shape of the aggregates [12].

In addition, environmental effects, including climate change and changes in air temperature, can significantly affect the stiffness of the asphalt mixture. This effect was measured by [10], demonstrating that with an increase in temperature, the stiffness of the asphalt mixture and its modulus decrease [10]. This phenomenon explains that temperature can significantly influence the rolling noise absorption of a pavement. In the literature, it has been reported that temperature changes can impact the noise generation in asphalt pavements [13,14]. However, the effect of temperature on the sound absorption of asphalt mixtures was unclear, and therefore the study of the sound absorption levels of specimens at various temperatures is fundamental.

The increase in temperature at the pavement surface can rapidly transfer heat to the tire tread, making the two stiffnesses similar to softening. Additionally, this aspect influences the reduction of noise generated by tire vibrations at frequencies between 1000 Hz and 2000 Hz [15]. Some authors have tried to show that the generated traffic noise and its propagation depend on the tire/pavement temperatures [16]. They recommend that noise measurements should not be made if the air temperature is below 15 °C or above 35 °C and propose a temperature correction of ±0.09 dB (A)/°C considering a reference air temperature of 20 °C [17–20] with noise being higher at 15 °C and lower at 35 °C. However, given this variability, the level of noise absorption at different temperatures is still unclear and further research is needed.

Other researchers have concentrated on the development of noise-reducing pavements utilizing crumb rubber (CR) obtained from end-of-life tires (ELTs) as an alternative and sustainable material, aiming to establish a quiet pavement variant using porous asphalt (PA). The addition of rubber in the form of aggregates in the porous–elastic pavement structure (PERS) achieves an attenuation of up to 8 dB (A) in the PA and 11 dB (A) in the PERS, according to the proximity methodology test (CPX) [17]. The attenuation achieved is relative to a dense pavement with a maximum aggregate size of 11 mm [21–23].

According to the literature, it is evident that the study of the effect of pavement temperature on noise pollution is of great interest to researchers [17,24,25]. This holds especially true for asphalt mixtures with rubber, as this material has been proposed as an alternative sustainable noise-attenuating additive in asphalt pavements.

Therefore, this research aims to evaluate the effect of recycled tire rubber at the end of its useful life as an alternative, sustainable, potential acoustic attenuator in asphalt mixtures, as well as the influence of temperature on this attenuation. For this purpose, the acoustic attenuation capacity has been evaluated by means of the absorption coefficient ($\alpha$). In addition, a fitting curve of Gaussian treatment has been made for the sound absorption measurement. To this purpose, several experimental spectra were investigated. The proposed approach allows the establishment of models relating the characteristics of asphalt mixtures to their sound absorption. To achieve these objectives, several asphalt mixtures were produced with different aggregate gradations and in various percentages of crumb rubber (CR) (0.0, 0.75, and 1.50%) by the dry method according to the total weight of the asphalt mixture. The acoustic performance of these mixtures was evaluated by conducting an experimental laboratory activity by means of an impedance tube, which made it possible to evaluate the sound absorption in different temperature conditions (10 °C to 60 °C). Finally, predictive models of the sound absorption spectra of the materials studied have been proposed.

## 2. Methodology and Materials

The methodology of this research is based on the comparison of the sound absorption of different categories of asphalt mixtures with and without the addition of crumb rubber (CR) and the proposal of predictive models of this absorption through the characteristics of the asphalt mixture and the temperature at which the absorption is measured. The CR was added by dry process (DP) technology as a fraction of the fine aggregate during the manufacture of the asphalt mixture. The CR has been incorporated by the dry process in proportions of 0.0% (reference mixtures), 0.75%, and 1.50% by weight of the total asphalt mixture. The asphalt mixtures are asphalt concrete (AC), stone mastic asphalt (SMA), and béton bitumineux mince (BBTM).

An experimental investigation was carried out as reported in Figure 1. Firstly, the asphalt mixture samples were manufactured in the laboratory and a characterization of their volumetric properties (air void contents, maximum and apparent densities) was carried out. Furthermore, non-destructive sound absorption tests were carried out using an impedance tube. A thermostatic chamber was used for temperature control. Lastly, the obtained acoustic results were treated by means of Gaussian fitting (GF) for each sound absorption spectrum, evaluating several characteristics such as Peak (P) shape, Area Under the Curve (AUC), and BandWidth (BW). This procedure was performed for five categories of asphalt mixtures, proposed in this study. The results obtained have been statistically treated, establishing models correlating the characteristics of asphalt mixtures and sound absorption. According to these models, the air void content of the mixture is the most influential factor affecting the absorption Peak, while the BandWidth and the Area Under the Curve of sound absorption depend on both the voids in the mixture and the CR content.

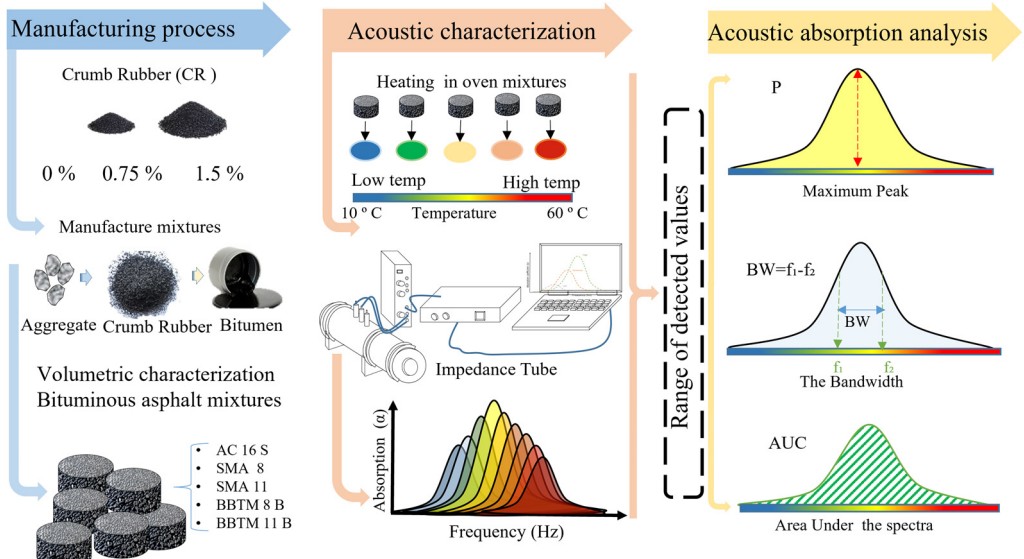

**Figure 1.** Methodology, including manufacturing process, evaluation, and analysis.

### 2.1. Sound Absorption

In order to evaluate the sound absorption and assess the contribution of CR as an attenuating material at different temperatures of asphalt mixture samples, a 4206 Bruel & Kjaer (B&K) impedance tube with frequency range of 100–1600 Hz was used. In this study, standard normal incidence sound absorption spectra were measured on asphalt mix samples with different crumb rubber contents and at different temperatures. Prior to the acoustic evaluation, the samples were laterally coated with a thin Teflon film (polytetrafluoroethylene—PTFE), as shown in Figure 2, to eliminate the clearance between the sample contour and the tube walls [26]. The EN ISO 10534-1 [27] and EN ISO 10534-2 [28] standards describe the impedance tube method and the transfer function technique (FFT). The signal emission equipment consists of the PULSE multianalyzer system type 3560 B-T06, and a B&K 2716C amplifier. The two

4187 microphones [29] are placed along the length of the tube, which allows the sound wave signals to be received. By means of the transfer functions (FFT), the sound absorption coefficient ($\alpha$) is determined with a frequency resolution of 1 Hz from a total of 100 averages per sound absorption spectrum at the microphone positions [29].

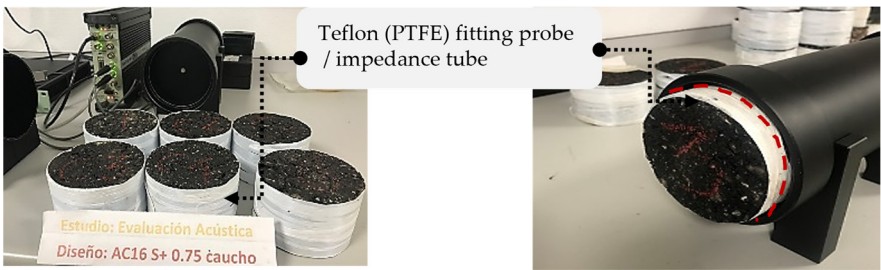

**Figure 2.** Impedance tube and specimens of asphalt mixtures with PTFE on their lateral surface.

## 2.2. Sound Absorption Analysis by Gaussian Adjustment

Using the Gaussian fit (GF), three parameters were determined from the acoustic response spectrum: Peak (P), BandWidth (BW), and Area Under the Curve (AUC). Based on the parameters and the response spectrum analysis techniques, it is possible to characterize the absorption by GF [30]. To explore the characteristics of the acoustic absorption spectrum, Gaussian fits have been successfully used in the analysis of the acoustic response of various materials [31,32]. All the fittings described in this subsection were carried out using the software Origin 8.0. The values of these parameters (P, BW, and AUC) are based on the general shape of the bell-shaped Gaussian function (Figure 3).

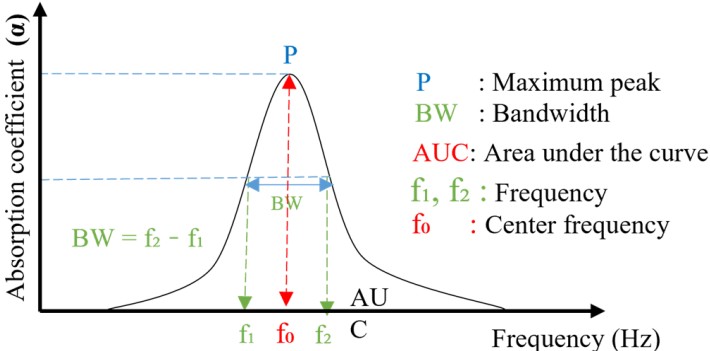

**Figure 3.** Geometric parameters for the characterization of the Gaussian fit.

In Figure 4, graph (a) shows a plot of the measured acoustic spectra for three types of asphalt mixtures (AC, SMA, and BBTM) and graph (b) shows the corresponding schemes of the parameters of the Gaussian fit: the fitted maximum Peak (P), the fitted BW, and the fitted AUC for each type of asphalt mixture. The values characterized by the parameters of the Gaussian fit were used to build correlation models with different volumetric characteristics of the asphalt mixtures, such as air void content, densities, permeability, and average depth of the surface texture. This analysis was established from consecutive and statistically representative measurements for the different pavements built with the asphalt mixtures studied.

Higher Gaussian adjustment parameters (P, BW, and AUC) are favorable for sound absorption. The P is related to the maximum sound absorption. The BW parameter could be related to sound absorption frequencies, as wider BandWidth allows a wider frequency range of sound absorption. The AUC parameter seems to most likely relate to the volumetric parameters, as its larger area would denote higher efficiency of the sound wave absorption process through the interconnected voids in the asphalt mixture.

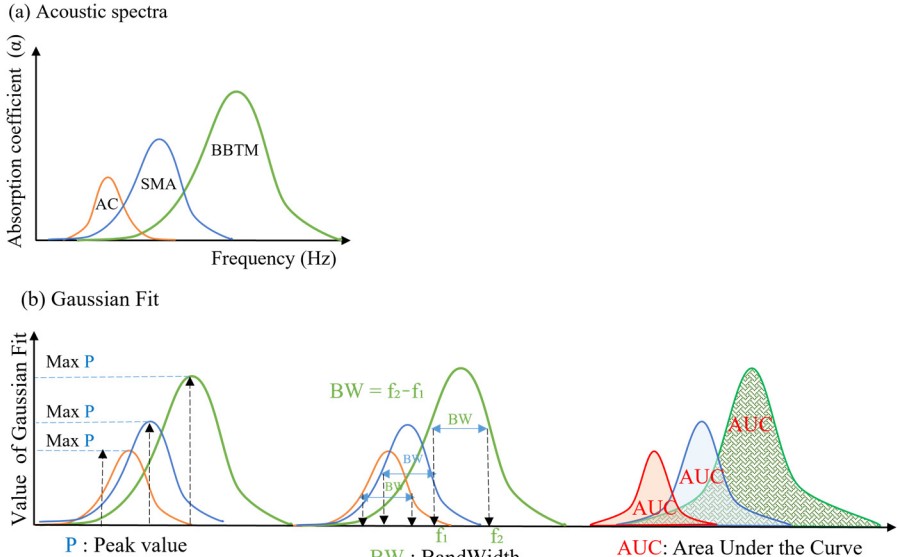

**Figure 4. (a)** Schematic of sound absorption spectra (ABS) for three types of asphalt mixtures, **(b)** Schematic of the Gaussian fits describing the ABS for three types of mixtures.

### 2.3. Determination of Volumetric and Permeability Characteristics

Volumetric characteristics are properties measured after the compaction procedure of asphalt mixtures. Density comprises two variants, (a) apparent density ($G_{mb}$) (g/cm$^3$) (EN 12697-6) [33] and (b) maximum density $G_{mm}$ (g/cm$^3$) (EN-12697-5) [34]. The $G_{mb}$ is the compacted density, including air voids obtained according to EN 12697-8 [33]. The $G_{mm}$ is a maximum value of the density that the mixture would have if it did not have air voids, and it is obtained by the pycnometer method and is calculated according to EN 12697-5 [34]. Together, $G_{mb}$ and $G_{mm}$ allow the determination of the air void content $V_a$ in (%) of a compacted asphalt mixture by the following expression:

$$V_a = \frac{G_{mm} - G_{mb}}{G_{mm}} * 100 \tag{1}$$

where: $G_{mm}$ is the maximum density in g/cm$^3$, $G_{mb}$ is the compacted apparent density in g/cm$^3$.

Macrotexture is measured by the mean texture depth (MTD) through the volumetric method of the sand patch test in mm. The procedure used for this work is based on EN 13036-1 [35] adapted to laboratory specimens. This method explains that the coarser the texture, the smaller the circle that can be covered by spreading the standard amount of sand. The calculation of the MTD value is expressed in Equation (2).

$$\text{MTD} = \frac{4V}{\pi \cdot D^2 avg} \tag{2}$$

where: V is the volume of the glass spheres in mL, D$avg$ is the mean diameter of the sand stain in mm.

The vertical permeability of asphalt mixture specimens was determined using the standard EN 12697-19 [36]. This method measures the permeability in the vertical direction by means of the time of drainage of a known volume of water through an asphalt sample. It determines the interconnection of internal voids in draining asphalt mixtures using Equations (3) and (4).

$$Q_v = \frac{(m_2 - m_1)}{t} \times 10^{-6} \tag{3}$$

where: $Q_v$ is the flow rate through the test tube (m$^3$/s); $m_1$ is the initial mass of the water on the sample (g); $m_2$ is the final mass of water (g); $t$ is the time of water drainage (s).

$$K_v = \frac{4 \cdot Qv \cdot L}{(h \cdot \pi \, D^2)} \tag{4}$$

where: $K_v$ is the vertical permeability (m/s), $Q_v$ is the vertical flow through the test sample (m$^3$/s), $L$ is the thickness of the sample (m), h is the height of the water (m), and $D$ is the diameter of the sample (m).

### 2.4. Statistical Analysis

A statistical analysis was performed to evaluate the effect of temperature, CR content, and volumetric properties of the asphalt mixtures (independent variables) on the Gaussian fitting parameters of sound absorption, i.e., P, BW, and AUC.

Prior to this, a correlation analysis was carried out to check for possible links between the independent variables, which could give rise to multicollinearity problems.

The Pearson correlation coefficient (*r*) has a range between [−1 and 1] [37]. The higher the absolute value of the coefficient *r*, the stronger the relationship between the two variables studied will be. Conversely, with values close to 0, the weaker the association between the two variables will be. The Pearson correlation coefficient is calculated using Equation (5).

$$r = \frac{\sum_{i=1}^{n} (x_i - \overline{x})(y_i - \overline{y})}{\sqrt{\sum_{i=1}^{n}(x_i - \overline{x})^2}\sqrt{\sum_{i=1}^{n}(y_i - \overline{y})^2}} \tag{5}$$

where: $n$ is the sample size, $x_i$ and $y_i$ are the ith sample points, and $\overline{x}$ and $\overline{y}$ are the sample means.

The degree of correlation between volumetric characteristics, CR content, and sound absorption measurement temperature was established according to Pearson's correlation grading rules (r) [37] which are shown in Table 1. The sign of the coefficient indicates the direction of the relationship (positive or negative). A positive value (r) indicates that as one variable grows so does the other, while a negative value (r) means the opposite trend.

**Table 1.** Degree of correlation by Pearson's coefficient.

| Classification Rules | Degree of Correlation |
|---|---|
| r = 0 | No correlation |
| 0 < ∣r∣ ≤ 0.19 | Very weak |
| 0.20 < ∣r∣ ≤ 0.39 | Weak |
| 0.40 < ∣r∣ ≤ 0.59 | Moderate |
| 0.60 < ∣r∣ ≤ 0.79 | Strong |
| 0.80 < ∣r∣ ≤ 1.00 | Very strong |
| ∣r∣ = 1 | Monotonic correlation |

On the other hand, the effects of volumetric characteristics, CR content, and temperature on the peak Gaussian fit parameters P, BW, and AUC obtained from the acoustic spectra have been evaluated by means of a multiple linear regression (MLR) analysis. Linear models have been estimated to predict the coefficients of the Gaussian fit P, BW, and AUC. The multiple linear regression model can be expressed as (Equation (6)):

$$y = \beta_0 + \beta_1 x_1 + \beta_2 x_2 + \cdots + \beta_n x_n \tag{6}$$

where: $y$ is the dependent variable (in this case P, BW, or AUC), $x_1$ to $x_n$ are the independent or predictor variables (in this case CR content, temperature, void content, average texture depth, permeability, and apparent density), $\beta_0$ is the $y$-intercept (constant term), and $\beta_1$ to $\beta_n$ are the regression coefficients estimated to improve the model fit.

Air void content $V_a$, CR content (%), and temperature (°C) were considered as potential predictors of Peak (P), BandWidth (BW), or Area Under the Curve (AUC) models, and the ordinary least squares (OLS) regression technique was used to estimate the regression coefficients. It is noteworthy that the type of mixture was not considered as a potential predictor, since it is believed that the main differences among the mixtures can be captured by variations in air void contents. The model estimation was carried out using "Statmodels", a Python library built specifically for statistical computing. A backward stepwise regression approach was used, with a significance level of α = 0.01. First, all potential predictor variables were included in the models excluding those that showed collinearity. Next, non-significant predictors were excluded from the models and the regression analysis was repeated. The procedure was stopped when the optimal model was identified after selecting the most appropriate and significant variables. Furthermore, the goodness of fit of the models was assessed by the coefficient of determination ($R^2$).

*2.5. Aggregates*

The aggregates used in this research are porphyry (of magmatic origin), generally used as a paving material for wearing courses in Spain. Due to the crushing process, the aggregate has an angular shape, which allows for a good mineral skeleton. The gradation of the aggregates is based on particle size curves according to articles 542, 543, and 544 of PG-3 [38]. The grading curves of the crushed aggregate of the five different mixtures (AC, SMA, and BBTM) are shown in Figure 5.

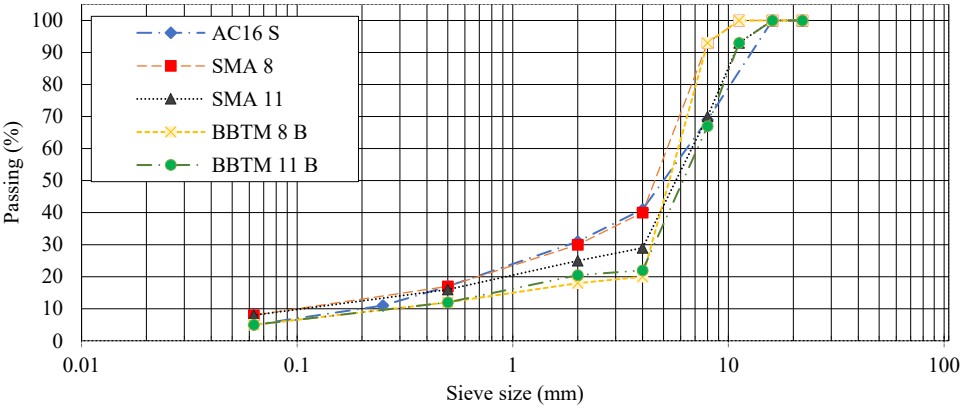

**Figure 5.** Aggregate gradation curves used in asphalt mixture design.

*2.6. Crumb Rubber (CR)*

Crumb rubber (CR) has been widely used in asphalt mixtures in several countries all over the world. For this research, a maximum particle size of 0.6 mm has been used. Table 2 presents the gradation curve of CR (EN 933-1, 2013). In this study, CR contents of 0% (reference), 0.75%, and 1.50% of the total weight of the asphalt mixtures were used.

**Table 2.** Gradation curve of Crumb Rubber (CR).

| Sieve (mm) EN 933-1-13 | 2 | 1.5 | 1 | 0.5 | 0.25 | 0.125 | 0.063 |
|---|---|---|---|---|---|---|---|
| Pass (%) | 100 | 100 | 100 | 94.1 | 23.7 | 3.7 | 0.4 |

*2.7. Asphalt Mixture Investigation: Laboratory Experimental Activity*

In this research, five asphalt mixtures, and more specifically AC 16 S, SMA 8, SMA 11, BBTM and BBTM 11 B, were manufactured with three different CR contents: 0%, 0.75%, and 1.50%. All mixtures were initially defined on the basis of a previous research work that demonstrated adequate mechanical behavior. For each mixture, six compacted specimens were manufactured, according to the Marshall EN 12697-34 method [39]. A

total of 90 specimens (5 mixtures for each of the 3 rubber contents for 6 specimens) were implemented and tested for the study.

Table 3 shows the manufacturing characteristics of the different asphalt mixtures studied in this research. The type of bitumen used is 50/70 asphalt binder according to EN 1426-07 [40]. This binder has been used because it has a good affinity with rubber, as widely demonstrated by previous works [26,41,42]. The samples were manufactured in the laboratory in a cylindrical shape of 60 mm in height and 99.5 mm in diameter to facilitate their accommodation in the impedance tube, which has an inner diameter of 100 mm.

**Table 3.** Manufacturing parameters.

| Mixture | Max. Size (mm) | Type of Mixture | CR (%) | Binder Content (%) | MT (°C) | Digestion Temperature (°C) |
|---|---|---|---|---|---|---|
| AC 16 S | 16 | AC 16 S—REF<br>AC 16 S—0.75%<br>AC 16 S—1.50% | 0.00<br>0.75<br>1.50 | 5.1<br>5.3<br>5.5 | 170 | 160 |
| SMA 8 | 8 | SMA 8—REF<br>SMA 8—0.75%<br>SMA 8—1.50% | 0.00<br>0.75<br>1.50 | 6.0<br>6.1<br>6.2 | 170 | 160 |
| SMA 11 | 11.2 | SMA 11—REF<br>SMA 11—0.75%<br>SMA 11—1.50% | 0.00<br>0.75<br>1.50 | 6.0<br>6.1<br>6.2 | 170 | 160 |
| BBTM 8 B | 8 | BBTM 8 B—REF<br>BBTM 8 B—0.75%<br>BBTM 8 B—1.50% | 0.00<br>0.75<br>1.50 | 5.0<br>5.1<br>5.2 | 170 | 160 |
| BBTM 11 B | 11.2 | BBTM 11 B—REF<br>BBTM 11 B—0.75%<br>BBTM 11 B—1.50% | 0.00<br>0.75<br>1.50 | 5.0<br>5.1<br>5.2 | 170 | 160 |

Note: CR: Crumb Rubber. MT: Manufacturing Temperature.

The increase in binder viscosity with the addition of CR is a well-known effect [43]. The increase in viscosity is caused by the integration of the rubber into the mixture once it comes into contact with the bitumen at high temperatures, resulting in an absorption of the lighter fractions of the bitumen (digestion process). It is also due to the fact that the (EN 12697-30, 2007) rubber increases in volume. This effect leads to a reduction of the distance between the particles and an increase in viscosity of the binder. For this reason, the manufacturing temperature of the mixtures with rubber has been set at 170 °C to facilitate the digestion process and the mixing of aggregates, binder, and CR.

Regarding the manufacture temperature (MT) of the test specimens, the aggregates were heated to 170 °C, and mixed with the crumb rubber for 30 s. The bitumen was then added at 170 °C and mixed with the aggregates and CR for approximately 30 s. The filler was then incorporated into the mixture and mixed for 120 s. The samples are then thermally conditioned to start the oven digestion temperature (DT) process at 160 °C, prior to the compaction process. The oven digestion time allows the asphalt mixtures to be kept at a constant temperature for 90 min so that the crumb rubber can better integrate with the bitumen and ensure proper digestion. Finally, all samples are compacted at 160 °C in the Marshall impact compactor in accordance with EN 12697-30 with application of 75 blows/surface for AC mixtures and 50 blows/surface for SMA and BBTM mixtures. Table 3 details the manufacturing parameters.

*2.8. Thermal Conditioning for Absorption Measurement*

For the evaluation of the influence of temperature on the sound absorption coefficients, the asphalt mixture samples were thermally conditioned, and their temperature was monitored using a FLIR C2 thermal imaging camera. The average surface temperature of the sample was obtained with the FLIR tools analysis software. All samples were kept

at an initial temperature of 10 °C, then conditioned per sample group until the desired temperatures of 20, 30, 40, 50, and 60 °C were reached in the oven (Figure 6a). The thermal conditioning is prolonged by 8 h for each temperature step. The effects of different temperatures are controlled by ensuring that the specimens are thermally stabilized before starting the test and confirming that there are no substantial variations during the measurement. Figure 6b shows the temperature check of the specimens with a thermographic camera.

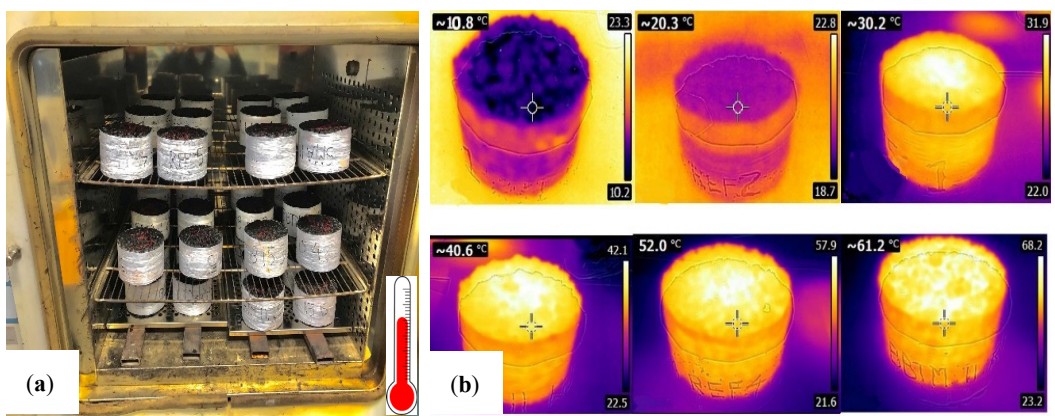

**Figure 6.** (**a**) Sample conditioning oven, (**b**) Thermographic imaging (10 °C to 60 °C).

## 3. Results and Discussion

### 3.1. Volumetric Characterization, Permeability, and Macrotexture

The volumetric parameters of the reference asphalt samples without CR and the mixtures with CR were determined in the laboratory. The maximum density ($G_{mm}$), apparent density ($G_{mb}$), air void content ($V_a$), hydraulic permeability ($K_v$), and mean texture depth (MTD) were obtained from the asphalt samples after compaction.

The volumetric characterization results for each type of asphalt mixture are shown in Table 4. From the results obtained, it can be seen that the reference mixtures (without CR) have a higher maximum density ($G_{mm}$) verified for all five types of asphalt mixtures, compared to the mixtures containing 0.75 and 1.50% CR. This is because the reference sample does not contain crumb rubber and its maximum density is related to the specific density of the aggregates and bitumen. However, the CR additions have a lower density than the aggregates. Therefore, the reduction in maximum density in the mixtures with CR is closely related to the presence of CR. This behavior of CR in asphalt mixtures was extensively studied by Gallego et al. [44].

The presence of CR slightly influences the air void content in AC asphalt mixtures. This could be due to its continuous particle size where the rubber can be better integrated into the mixture mass. However, rubber appears to increase voids in the SMA 8 and BBMT 8 B mixtures. This is probably due to the strong mineral skeleton of these two mixture types. It can be observed that the maximum aggregate size of 8 mm would be more affected by rubber particles in the compaction stage. The percentage increase in the voids with the addition of 1.50% CR was determined as follows: in AC16 it increases by 1.5% $V_a$, in SMA 8 by 38.4% $V_a$, SMA 11 by 1.6% $V_a$, in BBTM 8 by 10.40% $V_a$, and BBTM 11 by 0.79% $V_a$. The asphalt mixture exhibiting a significant increase in air void content is SMA 8 (38.4%). This is attributed to the expansion of the CR during the dry process of digestion. The lowest effect of CR is seen in BBTM 11 (0.79%). The presence of CR mainly impacted semidense mixtures with dimensions of the aggregate of 8 mm, while in the larger size of the aggregate, there was no significant effect of CR. This is due to internal porosity of open mixtures which allows the CR to swell and cover the internal pores without affecting it significantly. The presence of rubber in the SMA 11 and BBTM 11 B mixtures does not seem to generate as strong increases as in the SMA 8 and BBTM 8 B mixtures. This could be due to the size of the aggregate and the mineral skeleton it forms, which is smaller in the case of the 8 mixtures and more vulnerable to distortion by the presence of CR particles.

**Table 4.** Manufacturing detail, volumetric characteristic, permeability, and texture.

| Mixtures | CR (%) | MT (°C) | DT (°C) | $G_{mb}$ (g/cm³) | $G_{mm}$ (g/cm³) | $V_a$ (%) | $K_v$ (m/s) | MTD (mm) |
|---|---|---|---|---|---|---|---|---|
| AC 16 S | 0.00 | 160 | 170 | 2.406 | 2.540 | 5.27 | *- | 0.985 |
| | 0.75 | 160 | 170 | 2.358 | 2.487 | 5.24 | *- | 0.955 |
| | 1.50 | 160 | 170 | 2.295 | 2.424 | 5.35 | *- | 0.846 |
| SMA 8 | 0.00 | 160 | 170 | 2.385 | 2.507 | 4.86 | $4.41 \times 10^{-4}$ | 0.990 |
| | 0.75 | 160 | 170 | 2.342 | 2.483 | 5.67 | $7.53 \times 10^{-5}$ | 0.984 |
| | 1.50 | 160 | 170 | 2.296 | 2.461 | 6.73 | $7.18 \times 10^{-4}$ | 0.932 |
| SMA 11 | 0.00 | 160 | 170 | 2.349 | 2.517 | 6.66 | $1.97 \times 10^{-4}$ | 1.607 |
| | 0.75 | 160 | 170 | 2.316 | 2.483 | 6.73 | $9.26 \times 10^{-4}$ | 1.320 |
| | 1.50 | 160 | 170 | 2.252 | 2.442 | 7.78 | $6.34 \times 10^{-4}$ | 1.221 |
| BBTM 8 B | 0.00 | 160 | 170 | 2.210 | 2.650 | 16.61 | $2.98 \times 10^{-4}$ | 2.975 |
| | 0.75 | 160 | 170 | 2.136 | 2.591 | 17.56 | $1.98 \times 10^{-4}$ | 2.573 |
| | 1.50 | 160 | 170 | 2.117 | 2.592 | 18.34 | $1.93 \times 10^{-4}$ | 1.934 |
| BBTM 11 B | 0.00 | 160 | 170 | 2.148 | 2.643 | 18.72 | $2.79 \times 10^{-4}$ | 3.054 |
| | 0.75 | 160 | 170 | 2.140 | 2.634 | 18.76 | $3.05 \times 10^{-4}$ | 2.873 |
| | 1.50 | 160 | 170 | 2.135 | 2.632 | 18.87 | $3.03 \times 10^{-4}$ | 2.711 |

Note: CR: Crumb rubber, MT: Manufacturing temperature, DT: Digestion temperature, $G_{mb}$: Apparent density, $G_{mm}$: Maximum density, $V_a$: Air voids, $K_v$: Permeability, MTD: Mean texture depth, *-: Impermeable samples.

### 3.2. Sound Absorption Coefficient: Influence of Mixture Type, Temperature, and CR Content

In order to evaluate the influence of temperature on the sound absorption coefficient, several measurements were made at the following temperatures: 10 °C, 20 °C, 30 °C, 40 °C, 50 °C, and 60 °C ($\pm$2 °C). The sound absorption evaluations were quantified through the spectrum of the sound absorption coefficient ($\alpha$) in the samples previously conditioned at different temperatures. Figure 7 shows the sound absorption spectra for normal incidence at 20 °C for all AC, SMA, and BBTM asphalt mixtures with 0%, 0.75%, and 1.50% CR, at frequencies between 100 Hz and 1600 Hz. As shown in Figure 7, there is a strong difference in sound absorption at 20 °C between the different asphalt mixtures, with the BBTM mixtures having the highest sound absorption coefficient.

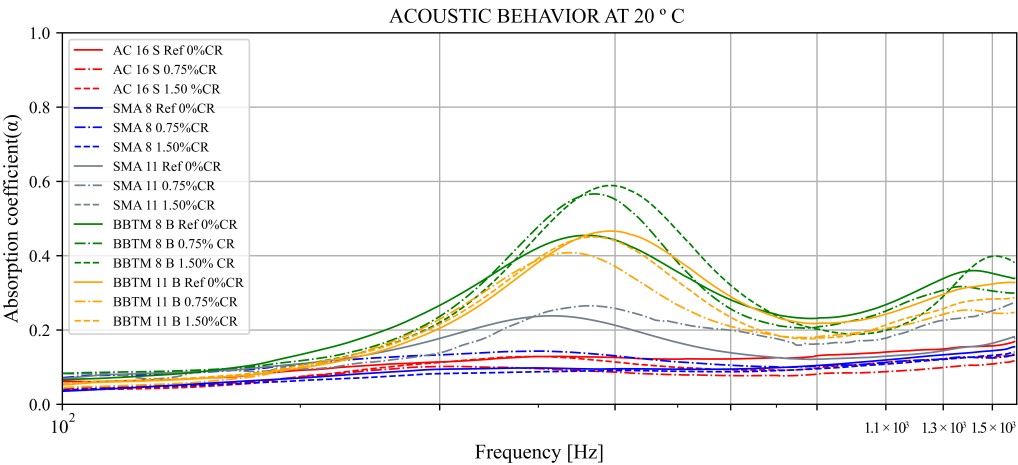

**Figure 7.** Sound absorption coefficient of the asphalt mixtures studied at 20 °C.

The behavior of the different sound absorption peaks might indicate that some asphalt samples show better sound absorption within a specific low-frequency range from 400 Hz to 700 Hz. Experimental studies from different investigations on asphalt mixtures showed similar sound absorption behavior at different frequencies [45,46], confirming that the peaks of maximum sound absorption are associated with the low-frequency range.

In addition, as shown in Figure 8, as the temperature increases, the levels of the maximum sound absorption coefficient also increase in the discontinuous BBTM mixtures. However, the changes in the maximum absorption coefficient in the dense AC and SMA mixtures due to the increasing effect of temperature are not evident.

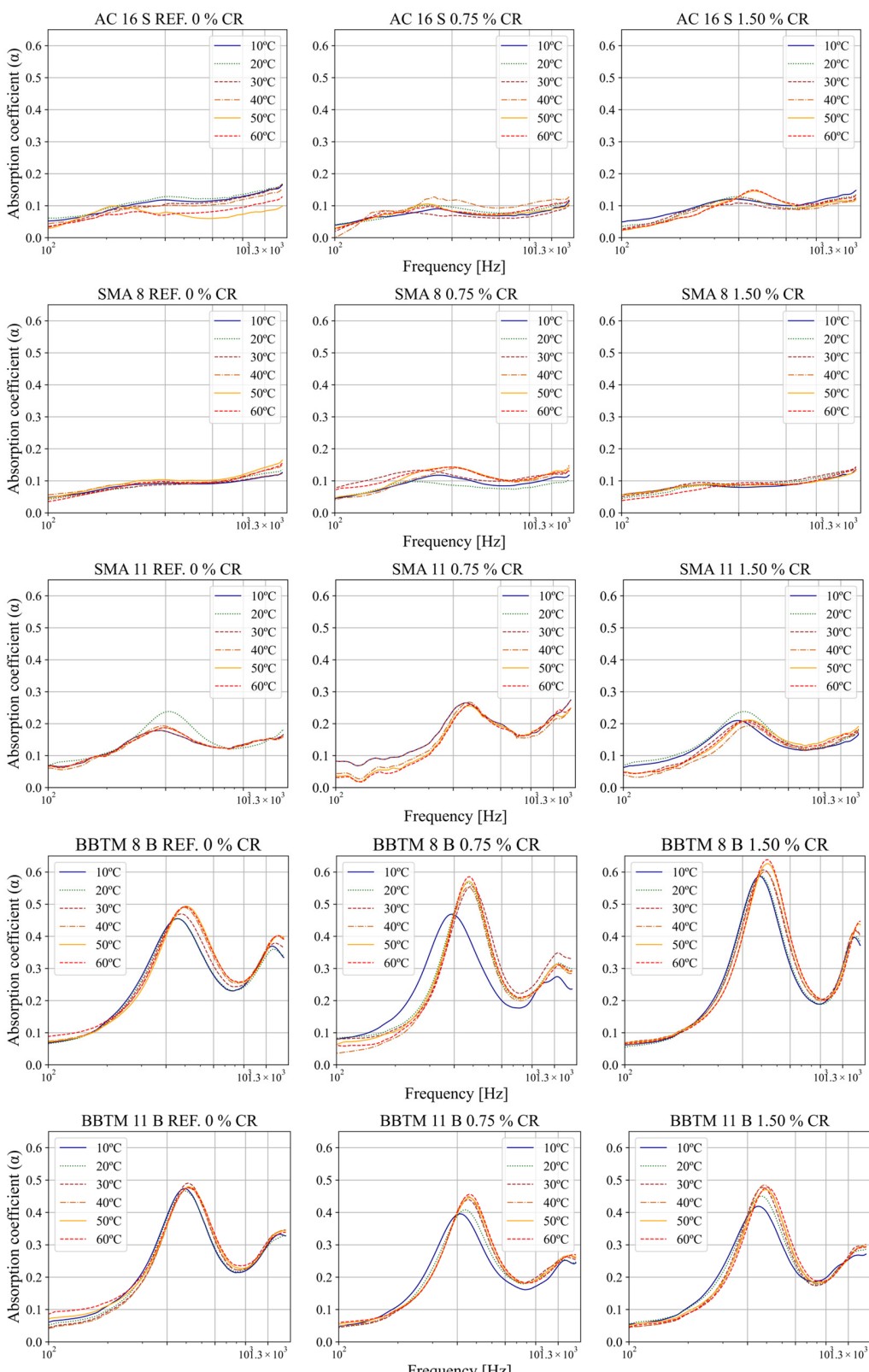

**Figure 8.** Sound absorption coefficient of the asphalt mixtures studied at different temperatures.

On the other hand, it is interesting to mention the behavior of the maximum sound absorption recorded in the BBTM mixtures, which show a positive behavior of the spectra and peaks of maximum sound absorption with slight shifts at higher frequencies. However, it is necessary to conclude that for practical purposes the measurement temperature has little influence, except for the BBTM mixture, where at 60 °C the sound absorption seems to be approximately 10% higher than at 10 °C. As for the CR content, it seems to have little influence on the sound absorption. The statistical significance of these variables will be analyzed in the following section.

### 3.3. Gaussian Goodness of Fit

Gaussian goodness of fit was assessed using the $R^2$ statistic (coefficient of determination) to check how well the normal distribution fits the observed data (sound absorption spectra). Figure 9 presents an example of this procedure applied for the BBTM 11 B with 1.50% CR mixtures, showing excellent goodness of fit results.

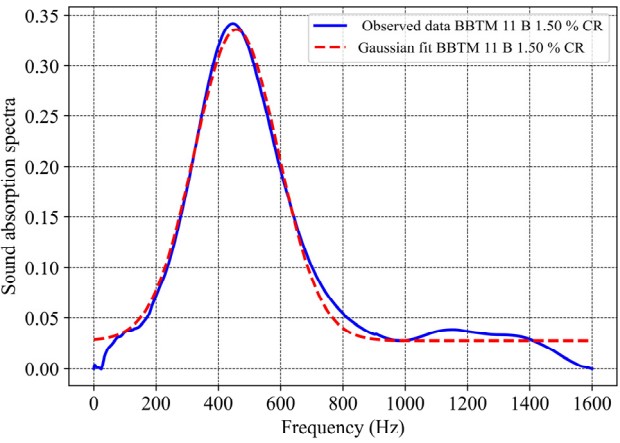

**Figure 9.** Gaussian fitting applicated for sound absorption spectra.

Table 5 illustrates the goodness of fit performance of all the mixtures under study, with values of $R^2$ always greater than 0.92.

**Table 5.** Result for the goodness of Gaussian fitting.

| Mixtures | CR (%) | No. Observations | $R^2$ | Adjusted $R^2$ | Result |
|---|---|---|---|---|---|
| | 0.00 | 801 | 0.923 | 0.923 | Accepted |
| AC 16 S | 0.75 | 801 | 0.966 | 0.966 | Accepted |
| | 1.50 | 801 | 0.943 | 0.943 | Accepted |
| | 0.00 | 801 | 0.986 | 0.984 | Accepted |
| SMA 8 | 0.75 | 801 | 0.922 | 0.922 | Accepted |
| | 1.50 | 801 | 0.984 | 0.985 | Accepted |
| | 0.00 | 801 | 0.924 | 0.924 | Accepted |
| SMA 11 | 0.75 | 801 | 0.941 | 0.940 | Accepted |
| | 1.50 | 801 | 0.963 | 0.962 | Accepted |
| | 0.00 | 801 | 0.969 | 0.968 | Accepted |
| BBTM 8 B | 0.75 | 801 | 0.958 | 0.958 | Accepted |
| | 1.50 | 801 | 0.967 | 0.966 | Accepted |
| | 0.00 | 801 | 0.993 | 0.931 | Accepted |
| BBTM 11 B | 0.75 | 801 | 0.989 | 0.989 | Accepted |
| | 1.50 | 801 | 0.981 | 0.980 | Accepted |

### 3.4. Regression Models of Asphalt Mixture Characteristics as a Function of Gaussian Parameters

The obtained results presented in the previous section have been analyzed through a statistical analysis. The first step consists in a correlation analysis, which is carried

out to investigate and assess potential correlations between the independent variables to prevent possible multicollinearity problems. To this purpose, the correlation between all potential predictor variables has been examined in order to determine the relations and to discard those that are correlated, thus avoiding multicollinearity problems in the predictive models of this research. Figure 10 shows the correlation matrix in which the relations between volumetric characteristics, temperature, and CR content, according to the Pearson coefficient, are shown.

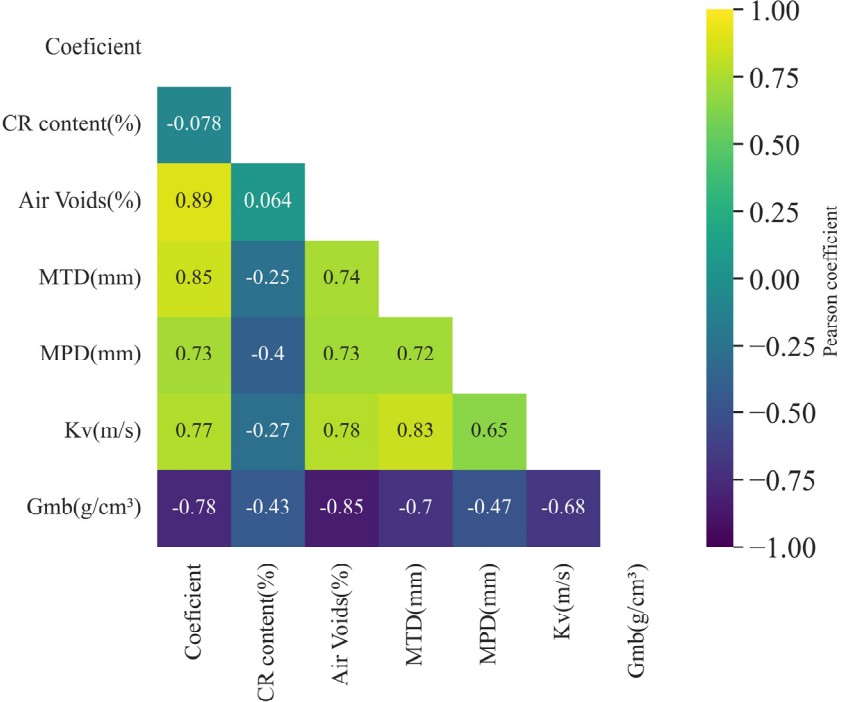

**Figure 10.** Pearson correlation analysis of temperature, rubber, and intrinsic parameters of asphalt mixtures.

The volumetric characteristics of the asphalt mixtures (AC16, SMA 8, SMA 11, BBTM 8 b, and BBTM 11 B) taken into consideration are: air void content ($V_a$), hydraulic permeability ($K_v$), mean texture depth (MTD), and the apparent density of the mixtures ($G_{mb}$). The maximum density $G_{mm}$ has not been considered, as it depends on the void content ($V_a$) and the apparent density ($G_{mb}$) by Equation (1). The volumetric characteristics of the mixtures have been found to be related to each other. However, the rubber content is only moderately related to $G_{mb}$.

A high degree of correlation between the volumetric properties of the mixtures has thus been found (Figure 10). More specifically, strong correlations have been found between the $V_a$ and the MTD, the $K_v$, and the apparent density $G_{mb}$. It is believed that the phenomenon of sound absorption is mainly due to the interconnected air voids within the mixture structure. For this reason, it has been decided to keep the air void variable $V_a$ in the predictive models and to remove MTD, $K_v$, and apparent density $G_{mb}$, in order to avoid collinearity and thus facilitate the interpretation of the predictive models presented below.

Furthermore, it has been observed that CR content and temperature did not show any strong correlation with the other variables, therefore they have been included in the predictive models for P, BW, and AUC.

The regression models of the Gaussian parameters, i.e., Peak (P), BandWidth (BW), and Area Under the Curve (AUC), are presented as a function of the air void content ($V_a$), the rubber content (% CR), and the temperature (°C) at which the measurement is made. It should be noted that the other volumetric properties (maximum density ($G_{mm}$), apparent density ($G_{mb}$), hydraulic permeability ($K_v$), and mean texture depth (MTD)) were

not considered as potential predictors, as they exhibited a strong correlation with void content as observed in Figure 10.

The results obtained were pre-treated for all the asphalt mixtures in Table 5 with the Gaussian curve fitting detailed in Section 3.3. Several attempts were made for each model, including the potential predictors (void content ($V_a$), absorption ($\alpha$) measurement temperature, and CR content), Furthermore, the process was iterated to obtain optimized models with statistically significant predictors. Tables 6–8 show the degrees of significance of all predictor variables used for the prediction models. In the Peak (P) model it has been observed that the CR content and temperature variables do not show high significance, while, for the BandWidth (BW) and Area Under the Curve (AUC) models, only temperature does not show significance.

**Table 6.** Peak (P) model.

| | Coefficients | | Statistics | | 95% Confidence Interval for β | |
|---|---|---|---|---|---|---|
| | β | Standard Error | t | *p*-Value | Lower Bound | Upper Bound |
| Intercept | −0.0301 | 0.028 | 1.055 | 0.292 | −0.026 | 0.086 |
| Air voids (%) | 0.0708 | 0.002 | 44.611 | <0.001 | 0.068 | 0.074 |
| CR content (%) | −0.0061 | 0.015 | −0.396 | 0.693 | −0.036 | 0.024 |
| Temperature (°C) | 0.0005 | 0.001 | 0.919 | 0.358 | −0.001 | 0.002 |
| No. observations | 540 | | | | | |
| R$^2$ | 0.788 | | | | | |
| Adjusted R$^2$ | 0.787 | | | | | |

R$^2$: Correlation coefficient, β: Estimated correlations coefficient, t: *t*-Student, *p*-value: Probability value.

**Table 7.** BandWidth (BW) model.

| | Coefficients | | Statistics | | 95% Confidence Interval for β | |
|---|---|---|---|---|---|---|
| | β | Standard Error | t | *p*-Value | Lower Bound | Upper Bound |
| Intercept | 416.770 | 11.341 | 36.748 | <0.001 | 394.495 | 439.053 |
| Air voids (%) | −10.650 | 0.811 | −13.126 | <0.001 | −12.242 | −9.055 |
| CR content (%) | 79.450 | 7.864 | 10.103 | <0.001 | 64.004 | 94.900 |
| Temperature (°C) | −0.352 | | −1.180 | 0.239 | −0.941 | 0.235 |
| No. observations | 540 | | | | | |
| R$^2$ | 0.327 | | | | | |
| Adjusted R$^2$ | 0.323 | | | | | |

R$^2$: Correlation coefficient, β: Estimated correlations coefficient, t: *t*-Student, *p*-value: Probability value.

**Table 8.** Area Under the Curve (AUC) model.

| | Coefficients | | Statistics | | 95% Confidence Interval for β | |
|---|---|---|---|---|---|---|
| | β | Standard Error | t | *p*-Value | Lower Bound | Upper Bound |
| Intercept | 18.504 | 2.931 | 6.314 | <0.001 | 12.747 | 24.261 |
| Air voids (%) | 8.157 | 0.210 | 38.907 | <0.001 | 7.745 | 8.569 |
| CR content (%) | −5.921 | 2.032 | −2.914 | 0.004 | −9.913 | −1.929 |
| Temperature (°C) | 0.1338 | 0.077 | 1.734 | 0.084 | −0.018 | 0.285 |
| No. observations | 540 | | | | | |
| R$^2$ | 0.740 | | | | | |
| Adjusted R$^2$ | 0.738 | | | | | |

R$^2$: Correlation coefficient, β: Estimated correlations coefficient, t: *t*-Student, *p*-value: Probability value.

Once the significant predictor variables have been determined (see Section 2.4), the results of the optimized models for Peak (P), BandWidth (BW), and Area Under the Curve (AUC) based on the 540 data set (5 mixtures × 3 CR contents × 6 measurement temperatures × 6 specimens) were obtained and are shown in Tables 9–11, respectively.

**Table 9.** Statistical parameters of the Peak (P) model.

| | Coefficients | | Statistics | | 95% Confidence Interval for β | |
|---|---|---|---|---|---|---|
| | β | Standard Error | t | *p*-Value | Lower Bound | Upper Bound |
| Intercept | 0.0424 | 0.020 | 2.164 | 0.031 | −0.004 | 0.081 |
| Air voids (%) | 0.0708 | 0.002 | 44.720 | <0.001 | 0.068 | 0.074 |
| No. observations | 540 | | | | | |
| $R^2$ | 0.788 | | | | | |
| Adjusted $R^2$ | 0.788 | | | | | |

$R^2$: Correlation coefficient, β: Estimated correlations coefficient, t: *t*-Student, *p*-value: Probability value.

**Table 10.** Statistical parameters of the BandWidth (BW) model.

| | Coefficients | | Statistics | | 95% Confidence Interval for β | |
|---|---|---|---|---|---|---|
| | β | Standard Error | t | *p*-Value | Lower Bound | Upper Bound |
| Intercept | 416.77 | 11.341 | 36.748 | <0.001 | 394.495 | 439.053 |
| Air voids (%) | −10.65 | 0.811 | −13.126 | <0.001 | −12.242 | −9.055 |
| CR content (%) | 79.45 | 7.864 | 10.103 | <0.001 | 64.004 | 94.900 |
| No. observations | 540 | | | | | |
| $R^2$ | 0.325 | | | | | |
| Adjusted $R^2$ | 0.322 | | | | | |

$R^2$: Correlation coefficient, β: Estimated correlations coefficient, t: *t*-Student, *p*-value: Probability value.

**Table 11.** Statistical parameters of the Area Under the Curve (AUC) model.

| | Coefficients | | Statistics | | 95% Confidence Interval for β | |
|---|---|---|---|---|---|---|
| | β | Standard Error | t | *p*-Value | Lower Bound | Upper Bound |
| Intercept | 18.504 | 2.931 | 6.314 | <0.001 | 12.747 | 24.261 |
| Air voids (%) | 8.157 | 0.210 | 38.907 | <0.001 | 7.745 | 8.569 |
| CR content (%) | −5.921 | 2.032 | −2.914 | 0.004 | −9.913 | −1.929 |
| No. observations | 540 | | | | | |
| $R^2$ | 0.738 | | | | | |
| Adjusted $R^2$ | 0.737 | | | | | |

$R^2$: Correlation coefficient, β: Estimated correlations coefficient, t: *t*-Student, *p*-value: Probability value.

In the Peak (P) model in Table 6, the variables temperature and rubber content (CR) were found to be non-significant and were therefore excluded from the model construction (Table 9). The model has a coefficient of determination $R^2 = 0.788$. The only significant variable was air void content ($V_a$), and the coefficient is positive, meaning that an increase in air voids in the asphalt mixtures leads to a higher Peak (P) sound absorption. More specifically, each one unit change in the air void content of the mixtures corresponds to an increase in Peak (P) by a factor of 0.0708.

The fitted line and the relationship between air void content ($V_a$) and Peak (P) are shown in Figure 11, and the model can be expressed as Equation (7):

$$P = 0.0424 + 0.0708 \cdot (V_a) \tag{7}$$

It should be noted that the model obtained is valid in the range of air void content considered in this study ($4\% < V_a < 19\%$).

In the BandWidth (BW) regression model, two significant variables were identified, $V_a$ and CR content (%). On the other hand, the temperature of the mixtures was found to be non-significant as seen in Table 6 and was therefore excluded from the model in Table 10. The coefficient of determination $R^2$ of the model was quite low, equal to 0.325, which means that the model can only explain 32.5% of the BW variability. The sign of the air void content ($V_a$) was negative (β = −10.648), which means that an increase in air voids

produces a decrease in the BandWidth (BW). In other words, a narrower BW concentrates more sound energy absorption in the frequencies that are most easily attenuated in the network of tunnels formed by the interconnected voids.

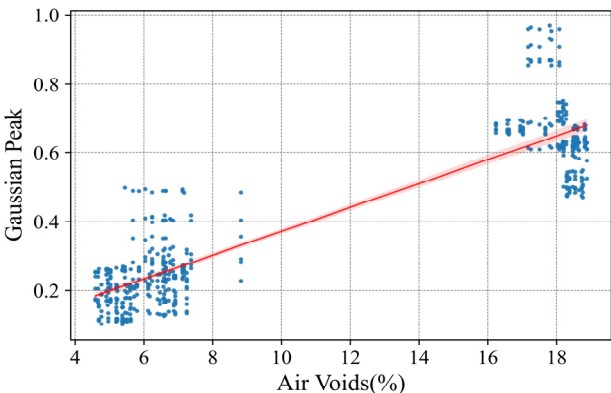

**Figure 11.** Linear correlation analysis of void content ($V_a$) and Peak (P).

On the other hand, the rubber content (CR) showed a positive coefficient, which means that the addition of CR tends to increase the BW. That is, it widens the range of frequencies of sound waves that are absorbed. Since the coefficient is high ($\beta = 79.452$) the effect is noticeable despite the small range of variation in crumb rubber content.

Figure 12 shows a 3D visualization of the multiple regression model for a better interpretation of the relationship between air voids, CR content, and BW. This model is within the limits of the void contents studied in this project ($V_a > 5\%$ and $V_a < 19\%$). This relationship can be expressed as:

$$BW = 416.77 - 10.648 \cdot V_a\ (\%) + 79.452 \cdot CR\ content\ (\%) \tag{8}$$

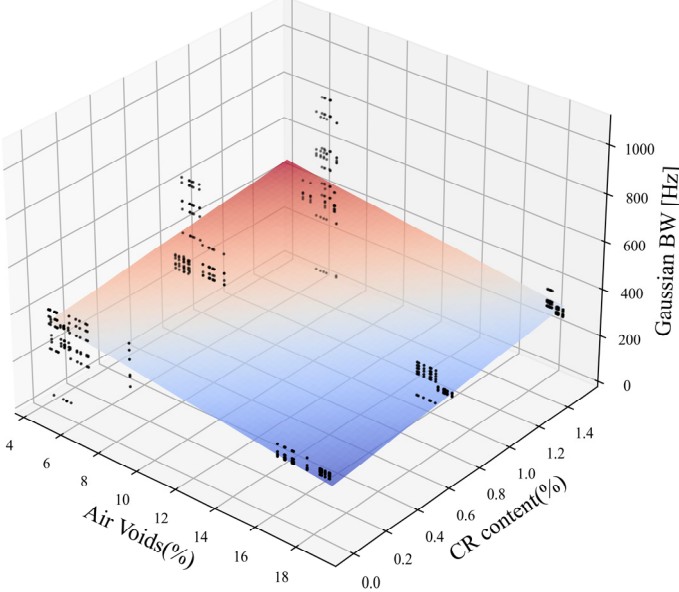

**Figure 12.** Multiple linear correlation analysis of void content, CR content, and BandWidth (BW).

Finally, in the Area Under the Curve (AUC) model, two significant variables, $V_a$ and CR content, were found. Again, temperature appears not to affect the acoustic response of the asphalt mixture (Table 8), is not significant, and is excluded in Table 11 of the optimized model. The coefficient of determination $R^2$ of the model was equal to 0.74, representing a strong relationship between the predictors and the dependent variable AUC. The coefficient of the air void content ($V_a$) was positive, meaning that an increase in air voids in the mixture

results in an increase in the AUC. Each one-unit change corresponds to an increase in AUC by a factor of 8.157, when the other predictor variable (CR content) is held constant [47–52].

On the other hand, CR content exhibited a negative coefficient, meaning that its effect is to reduce AUC.

Figure 13 shows the combined effect of air void content $V_a$ and CR content on the AUC. The model can be expressed as:

$$AUC = 18.504 + 8.157 \cdot V_a(\%) - 5.921 \cdot CR\ content(\%) \tag{9}$$

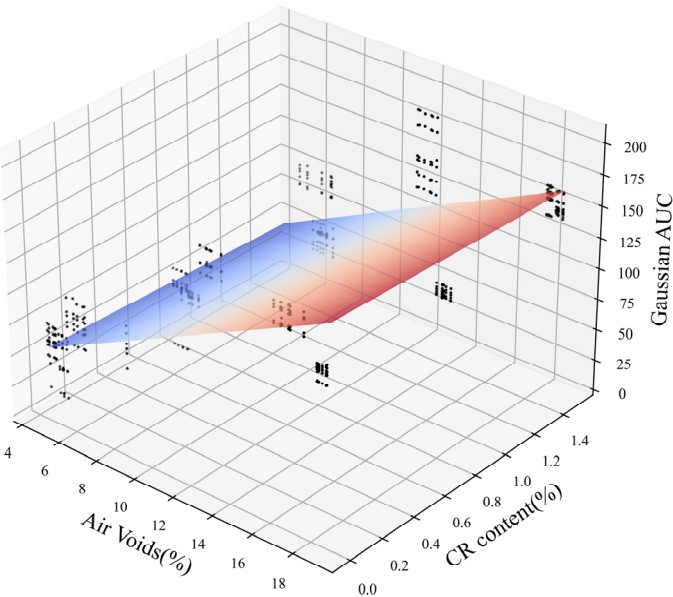

**Figure 13.** Multiple linear correlation between void content, CR content, and Area Under the Curve (AUC).

The results revealed that the Peak (P) of the Gaussian curve associated with the sound absorption is highly correlated and depends on the air voids ($V_a$) of the asphalt mixture. In other words, the results demonstrate that an increase in the porosity significantly improves the acoustic absorption performance. This finding aligns with the results obtained by other researchers [51]. More specifically, ref. [52] identified a linear relationship between the air voids of the mixture and the sound absorption coefficient with $R^2$ = 0.94.

Nevertheless, the Peak (P) does not stand as the unique variable delineating the acoustic response of asphalt mixtures. In addition, it was found that the CR had no significant effect on the Peak (P) sound absorption of asphalt mixtures. Experimental laboratory observations reveal that both air voids ($V_a$) and crumb rubber (CR) content exert discernible influences on the Area Under the Curve (AUC). The AUC serves as a comprehensive metric characterizing the acoustic response of asphalt mixtures across a spectrum of frequencies. Despite the proclivity of $V_a$ to positively correlate with AUC, indicating an augmentation in sound absorption properties, it is discernible that the introduction of CR marginally attenuates AUC. Corroborating this, Shatanawi et al. [47] have asserted that CR does not directly elicit an enhancement in sound absorption. Furthermore, the findings manifest that CR particles induce alterations in the volumetric properties of the mixtures, thereby influencing sound absorption characteristics.

The BandWidth (BW) provides information about the frequency-dependent variability of sound absorption properties. Mixtures without CR exhibit a tendency to concentrate absorption characteristics predominantly around the Peak, whereas with CR mixtures, the acoustic response manifests heightened stability surrounding the Peak. Furthermore, it is noteworthy that $V_a$ exhibits a reducing effect on the BW. Nonetheless, it is imperative to underscore the relatively low coefficient of determination for the model ($R^2$ = 0.325), indicative of its limited reliability. Subsequent investigations are warranted to advance

the comprehension of sound absorption properties inherent in sustainable road materials integrated into asphalt mixtures, with a specific focus on rubber. This entails the conduction of additional laboratory experiments and the exploration of materials aimed at amplifying the sustainability quotient of road construction materials.

## 4. Conclusions

The current research entails the assessment of sound absorption implementing an impedance acoustic tube for asphalt mixtures, both with and without crumb rubber (CR), across varied temperatures. The study encompasses dense asphalt concrete (AC), discontinuous stone matrix asphalt (SMA), and open-graded béton bitumineux mince (BBTM) formulations, with CR content of 0%, 0.75%, and 1.50% incorporated through the dry process (DP).

In-depth analyses of sound absorption spectra were conducted at diverse measurement temperatures ranging from 10 °C to 60 °C. Gaussian curve fitting was employed to extract parameters such as Peak (P), BandWidth (BW), and Area Under the Curve (AUC). The primary objective of this research is to explore potential correlations between the intrinsic characteristics of the asphalt mixtures and the Gaussian fit parameters derived from sound absorption results.

The establishment of prediction models for Gaussian absorption parameters, grounded in the distinctive features of the mixtures, allows the following conclusions:

- The Peak (P) absorption improves with the air void content ($V_a$) in asphalt mixtures. However, the rubber content (CR) and the temperature at which the measurement is made do not seem to influence the Peak parameter. The model fit is $R^2 = 0.788$.
- The air void content ($V_a$) tends to reduce the BandWidth (BW), while the crumb rubber content increases it. However, the model has limitations in terms of low goodness of fit, with $R^2 = 0.325$.
- Void content positively affects the Area Under the Curve (AUC). On the other hand, rubber content slightly reduces the AUC. This model has a good fit with a coefficient of $R^2 = 0.738$.

Based upon the finding of this research and in the context of practical applications, it can be asserted that:

- The main factor affecting the sound absorption is clearly the void content of the asphalt mixture. The maximum sound absorption coefficient was obtained for asphalt mixtures with high air void content, in this research the discontinuous BBTM open mixtures.
- Crumb rubber has limited influence on the sound absorption of AC and SMA mixtures. In open-mixture-type BBTM it seems to slightly improve the sound absorption.
- The temperature (10 °C to 60 °C) has a limited influence on the results.

The novelty of this research is the formulation of models describing the acoustic response of asphalt mixtures as a function of their volumetric characteristics and crumb rubber content. The results of this work constitute a useful starting point for the development and design of innovative sustainable sound-reducing pavements. However, there are other variables that need to be analyzed to improve the accuracy of the presented models and to gain a deeper understanding of this complex phenomenon. In this context, the effect of pavement aging, the nature of the aggregates, and the incorporation of different additives could be analyzed in future research. In addition, additional mixtures with a wider range of air voids should be studied. Future further analyses, including the realization of full-scale projects, would be crucial for the assessment of these technologies, which would allow for the mitigation of road noise pollution.

**Author Contributions:** Conceptualization, J.G.M., F.R.A. and V.F.V.; methodology, F.R.A., V.F.V. and J.G.M.; software, S.E.P.; validation, V.F.V., S.E.P. and V.G.; formal analysis, F.R.A., F.G. and V.G.; investigation, F.R.A.; resources, L.S.R.; data curation, V.F.V., F.G. and V.G.; writing—original draft preparation, F.R.A.; writing—review and editing, J.G.M., V.F.V., F.G. and V.G.; visualization, F.R.A., F.G. and V.G.; supervision, J.G.M.; project administration, S.E.P.; funding acquisition, L.S.R. All authors have read and agreed to the published version of the manuscript.

**Funding:** This work has been partially supported by the Ministry of Economy and Competitiveness in the project PID2020-118831RB-I00/AEI/10.13039/501100011033 in the framework of the National Plan for Scientific Research and by the Junta de Comunidades de Castilla-La Mancha (European Regional Development Funds—ERDF) in the project SBPLY/19/180501/000313. In addition, support and technical collaboration has been provided by Signus Ecovalor, a non-profit organization dedicated to the management of end-of-life tires in Spain.

**Data Availability Statement:** Data are contained within the article.

**Conflicts of Interest:** The authors declare no conflicts of interest.

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
