# Peer review of "Towards Sustainable Road Pavements: Sound Absorption in Rubber-Modified Asphalt Mixtures"

_infrastructures, doi:10.3390/infrastructures9040065_

Round 1
Reviewer 1 Report
Comments and Suggestions for Authors
1. Why study the sound absorption levels of specimens at various temperatures? After heating, the specimens will cool down to room temperature during testing, so what is the significance of the effects of various temperatures?
2. In Table 4, what do TF and DT represent in terms of temperature? This needs to be explained in the text.
3. From Figure 10, it can be seen that there is no strong correlation between CR content and various indicators of asphalt mixtures. CR mainly affects the mechanical behavior of asphalt mixtures rather than volumetric indicators. It is suggested that the author supplement some road performance indicators, such as rutting and low-temperature cracking.
4. In fact, the author needs to pay attention to an issue: it is generally believed that the role of CR in reducing noise on asphalt roads is due to the fact that CR reduces the elastic modulus of asphalt mixtures, which in turn reduces the impact between the vehicle and the tire, thereby reducing tire-road noise. On the other hand, porous asphalt mixtures, namely the BBTM used by the author (with a void ratio greater than 15%), reduce noise on the principle that air diffracts in the voids of the road surface structure, reducing the sound blasts emitted by air being squeezed out of tire patterns. Therefore, CR and porosity represent two completely different noise reduction technology routes. The standing wave tube method used by the author can only determine the noise reduction effect of the rich rough texture on the surface of the specimen and cannot measure the noise reduction effect brought by the reduced elastic modulus due to CR. However, as seen in Figure 7, specimens containing CR have better sound absorption effects. How does the author explain this?
5. The focus of the author's research is on noise reduction and sound absorption, but the author has paid too much attention to the volumetric parameters of asphalt mixtures and has not been able to clearly explain why some specimens have better noise reduction effects.
Comments on the Quality of English LanguageEnglish good.
Author Response
Dear reviewer, please find below the file which contains the responses to the comments received. Best regards

Reviewer 2 Report
Comments and Suggestions for Authors
This study looked into the sound absorption of asphalt pavement with recycled crumb rubber. Different types asphalt mixtures with three levels of rubber content were considered. The relationship among material volumetric properties, condition temperature, and sound absorption was investigated. The manuscript is well structured and presented. Detailed comments are as follows:
(1). Page 2, line 57, it is suggested to rephrase the sentence. It is unclear why the authors emphasize the relationship between air temperature and pavement temperature.
(2) Page 8, line 271, it would be beneficial for the authors to provide additional details about the crumb rubber (CR) content. Clarification on whether it is based on the binder or mixture is needed?
(3) line 331-340: it is recommended to make modifications to this paragraph. It seems like with the CR increase, most of these mixes experienced increased air void, not just SMA 8 and BBMT 8B. It is also recommended to calculate the percent increase of air void to better quantify which mixture has more impact from crumb rubber. For instance, the statement in line 335, "would be more affected," could be enhanced by calculating the percent increase.
(4) Line 472: is it supposed to be Va<19%?
Author Response

(The authors gave the same response as above.)

Reviewer 3 Report
Comments and Suggestions for Authors
The topic of the paper is a current issue in global research, as it conforms with the goals of pavement sustainability combined with technical issues, which arise during the service life of road pavements, namely traffic noise. Overall, the manuscript is well written and every step of the methodology is justified. However, there are a few issues, which needs to be explained or better determined in the text. Thus, the following comments occur:
· Why did the authors use the gaussian fitting?
· Acronyms, such as SMA and BBTM, should be explained when they first appeared in the text.
· In the footnotes of Table 3, MT may be TF instead.
· In line 362, why is the word “evident” in parenthesis?
· The section of “Results” should be renamed to “Results and “Discussion”.
· In line 389, the word “whit” may be “with” instead.
Author Response

(The authors gave the same response as above.)

Round 2
Reviewer 1 Report
Comments and Suggestions for Authors
I'm not sure why the journal did not receive the review comments I previously submitted. I am very satisfied with the current version; the author has made very detailed responses and revisions. Good work.
Comments on the Quality of English LanguageEnglish is good enough.
Author Response
Thank you for your comments.